# Current Model Systems for Investigating Epithelioid Haemangioendothelioma

**DOI:** 10.3390/cancers15113005

**Published:** 2023-05-31

**Authors:** Emily Neil, Valerie Kouskoff

**Affiliations:** School of Medical Sciences, Faculty of Biology, Medicine and Health, The University of Manchester, Manchester M13 9PT, UK

**Keywords:** EHE, TAZ-CAMTA1, model system, vascular cancer

## Abstract

**Simple Summary:**

Epithelioid haemangioendothelioma is a rare type of cancer with an unpredictable disease course and very few treatment options. To better understand how this type of cancer develops and to uncover possible lines of treatment, it is critical to have experimental approaches to study this cancer. Here, we describe and compare the model systems that are currently available to study this disease. The research undertaken and the discoveries made using each of these experimental models is presented, and the advantages and disadvantages of each model are discussed.

**Abstract:**

Epithelioid haemangioendothelioma (EHE) is a rare sarcoma of the vascular endothelium with an unpredictable disease course. EHE tumours can remain indolent for long period of time but may suddenly evolve into an aggressive disease with widespread metastases and a poor prognosis. Two mutually exclusive chromosomal translocations define EHE tumours, each involving one of the transcription co-factors TAZ and YAP. The TAZ-CAMTA1 fusion protein results from a t(1;3) translocation and is present in 90% of EHE tumours. The remaining 10% of EHE cases harbour a t(X;11) translocation, resulting in the YAP1-TFE3 (YT) fusion protein. Until recently, the lack of representative EHE models made it challenging to study the mechanisms by which these fusion proteins promote tumorigenesis. Here, we describe and compare the recently developed experimental approaches that are currently available for studying this cancer. After summarising the key findings obtained with each experimental approach, we discuss the advantages and limitations of these different model systems. Our survey of the current literature shows how each experimental approach can be utilised in different ways to improve our understanding of EHE initiation and progression. Ultimately, this should lead to better treatment options for patients.

## 1. Introduction

Dysregulation of the Hippo pathway is observed in a vast array of cancers, and is attributed to the development of multiple cancer cell characteristics [1,2]. These include increased cell proliferation, survival, and metastasis. Transcription factor with PDZ domain (TAZ; encoded by the *WWTR1* gene) and yes-associated protein 1 (YAP1) are the transcriptional co-activators downstream of the Hippo pathway which facilitate the onset of these characteristics [2,3,4,5]. This is achieved through initiation of a broad transcriptional programme upon interaction with transcription factors [3,6,7,8]. TAZ/YAP mutations in cancer occur almost exclusively by means of gene fusion, with point mutations being very rarely discovered [2,9]. The resultant fusion proteins promote an aberrant transcriptional programme, retaining key features of wild-type TAZ/YAP alongside those contributed by the fusion partner [9,10,11]. TAZ-CAMTA1 (TC) and YAP1-TFE3 (YT) are two examples of TAZ/YAP fusion proteins, which are present exclusively in epithelioid haemangioendothelioma (EHE) [12,13,14,15]. Functional investigations of TC and YT that have been carried out recently have provided insight into the oncogenic functions common across TAZ/YAP fusion proteins.

Until recently, the lack of representative EHE models made it challenging to study the mechanisms by which TC and YT promote tumorigenesis. This is a significant issue with rare diseases, such as EHE, due to the scarcity of patient samples. Here, we discuss currently available EHE model systems and how they have furthered our understanding of EHE biology. We also speculate on how these models could be used and further developed to uncover novel aspects of EHE biology.

## 2. Epithelioid Haemangioendothelioma: Clinical Characteristics and Molecular Alterations

EHE is a rare sarcoma which arises from vascular endothelial cells [12,16]. EHE has an unpredictable disease course, with cases ranging from relatively indolent to highly aggressive [12,17,18,19]. EHE tumours have been reported in various parts of the body, although typically involve the lungs, liver, or soft tissues [18,20,21]. The location of the EHE tumour has an influence on prognosis, as patients with lung or liver EHE typically have worse survival rates than those with soft tissue tumours [13,18,21]. The variability in the presentation and in the course of the disease makes EHE difficult to treat.

A recurrent t(1;3)(p36;q25) chromosomal translocation was first identified in EHE patient samples in 2001 [22]. This was later characterised as resulting in the expression of the TC fusion protein, which is present in up to 90% of EHE cases [15,23]. Since then, the YT fusion protein (t(X;11)(p11;q22) translocation) was discovered to be present in the remaining 10% of EHE tumours [14]. TAZ and YAP are the N-terminal proteins in TC and YT, respectively [14,15]. Hence, both fusion proteins retain the TEAD-binding domain and key LATS1/2 phosphorylation residues of wild-type TAZ/YAP, which are common characteristics of all known TAZ/YAP fusion proteins [9,11,24]. The C-terminal fusion partners of TC and YT, calmodulin-binding transcriptional activator 1 (CAMTA1) and transcription factor E3 (TFE3) are both transcription factors [25,26]. Little is known about CAMTA1; however, evidence suggests it may function as a tumour suppressor [27,28]. TFE3 is a member of the microphthalmia associated transcription factor (MiTF) family, which regulates cell stress responses and proliferation [26]. In addition to TC and YT, expression of the rare TAZ-ACTL6A and TAZ-MAML2 fusion proteins has been characterised in a subset of cardiac EHEs [29].

The known TAZ/YAP fusion proteins present in EHE are mutually exclusive of each other. In addition to the disease-defining TAZ/YAP fusion proteins, analysis of EHE patient samples have revealed that at least 50% harbour a secondary mutation [18,30]. The most common mutation causes inactivation of *CDKN2A*, which encodes key positive regulators of cell growth arrest and senescence [30,31]. Other commonly mutated genes were typically involved in regulation of cell growth or DNA damage repair [18,30]. These investigations suggest that the presence of a secondary mutation is correlated with increased tumour aggressiveness in EHE patients.

## 3. An Overview of the Hippo Pathway

Given that all EHE tumours are defined by expression of a TAZ/YAP fusion protein, it is evident that they are central to EHE oncogenesis [9,14,15,29]. Wild-type TAZ/YAP is implicated in the development of multiple cancer types, including melanoma and breast cancer [2,32,33,34,35,36]. Often this is due to dysregulation of Hippo signalling, the pathway which primarily negatively regulates wild-type TAZ/YAP [37]. Furthermore, oncogenic TAZ/YAP fusion proteins are typically less sensitive to Hippo pathway regulation [9,11,24,38,39]. The Hippo pathway is highly conserved between species, and is key for the regulation of cell proliferation, tissue regeneration and stem cell maintenance [3,40,41]. Loss-of-function of Hippo pathway components in *Drosophila* (Hpo; homolog of MST1/2) and Warts (Wts; homolog of LATS1/2) resulted in the overgrowth of imaginal discs and the related adult organs [40,42]. The canonical Hippo pathway consists of two core serine/threonine kinases, mammalian STE20-like protein kinase 1/2 (MST1/2) and large tumour suppressor homolog 1/2 (LATS1/2) [6,43,44]. The adaptor proteins Salvador homolog 1 (SAV1) and MOB kinase activator 1A/B (MOB1A/B) interact with LATS1/2 and MST1/2, respectively [45,46]. LATS1/2-mediated phosphorylation of TAZ/YAP allows their interaction with 14-3-3 proteins, resulting in cytoplasmic retention and inactivation. MAP4Ks and serine/threonine kinase 25 (STK25) have also been identified as being able to activate LATS1/2, working alongside MST1/2 [47,48].

TAZ/YAP must interact with other transcription factors as they do not contain a DNA binding domain themselves [4,5]. This is primarily achieved through interactions with TEAD-family transcription factors (TEAD1-4), with TAZ/YAP containing a specific N-terminal binding domain to this effect [8]. Examples of TAZ/YAP-TEAD target genes include connective tissue growth factor *(CTGF)* and cysteine rich angiogenic inducer 61 *(CYR61)* [3]. TAZ/YAP-TEAD can also interact with other transcriptional complexes, such as AP-1 and E2F [49,50]. Here, the complexes act synergistically to initiate expression of specific sets of YAP/TAZ-TEAD target genes. TAZ/YAP can also interact with multiple PPXY motif-containing transcription factors, through their WW domains [3]. Examples include SMAD2/3/4, RUNX2, and p73 [4,51,52,53]. Besides the canonical Hippo pathway, TAZ/YAP activity can be regulated by other means including the WNT and mTOR pathways, and the actin cytoskeleton [54,55,56,57,58,59]. Hence, TAZ/YAP are subject to a complex network of upstream regulation.

## 4. EHE Model Systems

In recent years, a range of model systems for investigating EHE have been reported, and together have greatly improved our understanding of the disease [10,38,39,60,61,62,63]. These include in vitro and in vivo systems established using different approaches (Table 1). Through these models, it has been possible to elucidate how TC and YT are regulated by the Hippo pathway, the transcriptional programme they control and the cellular consequences of their expression [10,38,39,61]. Some of these findings have also provided a rationale to begin clinical studies of potential EHE treatments [60]. Below, we discuss the benefits of each model system, and their contribution to the best of our knowledge of EHE biology.

### 4.1. Cell Line-Based Models

Seminal studies on the regulation and function of TC have made use of immortalised cell lines [10,38,60] (Figure 1). Cell line-based models for studying cancer have many benefits (Table 2). Typically, these cell lines require relatively simple culture conditions, which is both cost effective and makes it easy to establish their use. Another benefit is the ability to scale-up, meaning that sample availability is not prohibitive when performing downstream assays. As such, these have been employed to study the mechanism of action of many oncogenes, including other TAZ/YAP fusion proteins, besides TC and YT [9,11,64,65].

The first functional study into TC introduced its ectopic expression into human embryonic kidney (HEK293) and murine fibroblast (NIH3T3) cell lines, prior to analysis of cell proliferation and transcriptional landscape [38]. This was necessary, as at the time, there were no EHE cell lines available. This investigation uncovered that TC is less sensitive to negative regulation by the Hippo pathway, with mutation of the critical serine 89 (S89) LATS1/2 phosphorylation site having no effect on the colony forming ability of TC-expressing cells [38]. TC retained a predominantly nuclear localisation despite phosphorylation by LATS1/2, likely due to the C-terminal nuclear localisation signal (NLS) present in the CAMTA1 moiety. Consistent with this, analysis showed that TC had a nuclear localisation in EHE patient samples [13]. Later, Driskill et al. utilised HEK293T cells to reveal that TC still binds 14-3-3 proteins despite its nuclear localisation [39]. They also showed that TC could be regulated mechanically, as addition of the F-actin inhibitor latrunculin B enabled TC to switch to a more cytoplasmic localisation [39]. Regulation by mechanical signals is well described for wild-type TAZ/YAP, often independent of Hippo pathway signalling [55,59,66,67].

Another key finding of this study determined that TC initiates a predominantly ‘TAZ-like’ transcriptional programme, through its interaction with TEAD4 [38]. Further research using the same model system has shown that TC mediates the expression of *CTGF*, a known transcriptional target of wild-type TAZ/YAP [60]. CTGF protein was found to be critical for driving increased cell growth downstream of TC activity, via an integrin-Ras-MAPK signalling pathway. Another study built on the findings of Tanas et al., and characterised a role for the TAZ/YAP fusion partners in shaping the transcriptional landscape governed by TC or YT [10,38]. This study employed a similar model system of ectopically expressing TC or YT in NIH3T3 or human liposarcoma (SW872) cell lines. Here, RNA-sequencing confirmed that TC or YT mediates a transcriptional programme similar, but not identical, to the activated TAZ/YAP mutants TAZ4SA and YAP5SA [10]. This is in agreement with transcriptomic studies using alternative EHE model systems [11,38,39,61,62,63].

Further investigation revealed that TC and YT occupy both TEAD and non-TEAD transcription factor binding motifs, including early growth response gene 2 (EGR2) motifs for TC, and microphthalmia-associated transcription factor motif (MiTF) motifs for YT [10]. Moreover, the CAMTA1 and TFE3 moieties mediate recruitment of the ADA-2A containing histone acetyltransferase complex (ATAC), increasing chromatin accessibility at TC and YT target genes [10]. Hence, this study improved our understanding of the mechanisms underpinning TC and YT transcriptional activation, particularly by uncovering the contributions of CAMTA1 and TFE3. These findings also provide an important link between the mechanism of action of TC and YT and other sarcoma-defining fusion proteins, which often require interaction with a chromatin-modifying complex to exert their oncogenic function [68].

Another key advantage of cell line-based models is the ability to introduce xenografts into mouse models. This permits in vivo tumorigenesis and metastasis studies, while being less time and resource intensive than developing a GEMM. Merritt et al. took advantage of this, creating mouse xenograft models with the same TC or YT expressing NIH3T3 or SW872 cells used in in vitro experiments [10]. These models showed shorter latency to tumour development, with both TC and YT expressing more xenografts than empty vector controls. Additionally, histological analysis revealed microscopic metastases in the lungs of xenografted mice, a common metastatic site in human EHE [10,13,20]. Ma et al. also generated murine xenograft models to further characterise the requirement of Ras-MAPK signalling for driving TC-mediated tumorigenesis in vivo [60]. This enabled the efficacy of trametinib, a clinically available MAPK inhibitor, to be evaluated. Promising results were obtained from this investigation, providing rationale for a phase II clinical study into the use of trametinib for metastatic EHE treatment [60].

### 4.2. Stem Cell-Based Models

One drawback of the cell line-based models discussed above is the lack of contextual relevance to EHE, as they are not of endothelial origin. This is of particular importance, given the cell-context-specific functions of wild-type TAZ/YAP [41,69,70]. Attempts to express the TC fusion protein in primary endothelial cells, such as human umbilical vein endothelial cells (HUVECs), have been unsuccessful. Our work has addressed this issue by developing an EHE model whereby endothelial cells are derived from mouse embryonic stem cells (mESCs), an approach used successfully to study various other diseases [63,71,72] (Figure 2). The differentiation protocol entailed the generation of embryoid bodies (EBs) from mESCs, from which an endothelial progenitor population was isolated and re-plated for further culture [63]. Throughout the protocol, various growth factors and substrates were used to replicate vasculogenic and angiogenic processes, which guide in vivo endothelial development. This model system retains many benefits of the in vitro cell line-based models, while permitting study of previously unexplored aspects of EHE biology (Table 2).

This EHE model utilises a doxycycline-inducible system for inducing TC expression. This prevents TC from interfering with the differentiation process; however also permits the study of the immediate consequences of TC expression, which is not possible with other EHE models. Investigations using this model provided evidence that TC expression causes hypertranscription, leading to DNA damage in endothelial cells [63]. As homologous recombination (HR), the least error-prone DNA repair pathway, appears to be impaired in TC expressing endothelial cells, many undergo oncogene-induced senescence (OIS) [63,73,74]. It is thought that the onset of OIS could represent the indolent EHE tumours often present in patients. A similar mechanism of action has been reported for the EWS-FLI1 fusion protein found in Ewing sarcomas, providing more evidence for a functional overlap between distinct sarcoma-defining fusion proteins and TC [75].

Another benefit of this model was that secondary mutations could be introduced into the endothelial cells relatively easily, using CRISPR/Cas9 methods [63]. We uncovered that *Cdkn2a* deletion, the most common secondary mutation in EHE tumours, permitted senescence bypass and increased cell growth [18,30,63]. This is consistent with analysis of patient samples, which found EHE tumours harbouring secondary mutations to be more aggressive [18,30]. Moreover, this study compared endothelial populations expressing different levels of TC [63]. This revealed that, in many of the biological processes investigated, cells expressing high levels of the TC fusion protein displayed more severe alterations than cells expressing low levels of the fusion protein. Hence, it is possible that TC expression level plays a role in determining the severity of EHE disease in patients and warrants further investigation.

### 4.3. Genetically Engineered Mouse Models

In addition to the in vitro models described above, genetically engineered mouse models (GEMMs) of EHE have been developed (Figure 3). The availability of a GEMM is important for studying disease, as these permit functional investigation of an oncogene in the correct cellular context and can be used for drug testing. Moreover, evaluation of the tumour microenvironment and metastasis is possible in this type of model. Regarding EHE, different approaches have been successful at introducing the TC transgene into these GEMMs [39,61,62]. The first of these used a doxycycline inducible system, whereby TC expression was under the control of a minimal CMV promoter and tetracycline response element (TRE) [39]. Tetracycline transactivator (tTA) expression was under the control of the *Cdh5* promoter, resulting in a mouse model that expressed TC specifically in the endothelium. Furthermore, TC expression could be repressed upon doxycycline administration [39]. These mice developed tumours with the clinical characteristics of human EHE, including histological similarities and the location of metastases in the lung [12,13,39]. Moreover, repression of TC expression in the mice by administration of doxycycline caused tumour regression and a reduction in Ki67 expression [39]. This suggests that TC expression is required for both tumour initiation and maintenance. One limitation of this model is that TC expression is not controlled under the *WWTR1* promoter, as is the case in EHE [15,39].

A second GEMM overcame this limitation by using a flip-excision (FLEx) system and Cre recombinase [61]. The FLEx system has been used previously in the generation of transgenic mice; however, this study represents the first to apply it for studying a gene fusion. Cre expression was induced by tamoxifen, under the control of either the *Cdh5* or the *Rosa26* promoter to allow endothelial specific or ubiquitous expression, respectively [61]. This model also produced tumours histologically similar to human EHE when two copies of the TC transgene were present. The GEMM had a long latency to tumour development, representative of indolent disease in EHE patient; however this presents logistical challenges for downstream analysis. Both GEMMs uncovered that the TC transgene is embryonic lethal, and produced findings consistent with in vitro studies regarding the overlap between the TC and TAZ/YAP transcriptional programmes [10,38,39,61]. Seavey et al. used their GEMM to characterise a gene set specific to TC-driven EHE, which was found to be highly significantly enriched for in human EHE disease [61]. This gene set is extremely useful for confirming the relevance of EHE models to the human disease.

One GEMM was further developed to determine the contribution of *Cdkn2a* knockdown to EHE tumourigenesis [61,62]. Here, mice with Cre-mediated deletion of exons 2 and 3 of the *Cdkn2a* gene were crossed with those harbouring the TC transgene. This resulted in mice with endothelial-specific loss of functional p16^INK^ and p19^ARF^ protein expression, with TC expression induced by tamoxifen administration [61,62]. This revealed that loss of *Cdkn2a* caused reduced latency to tumour development and increased Ki67^+^ fraction, while maintaining the EHE specific transcriptome [62]. This is in line with studies to the same effect in vitro, and provides further evidence that a secondary mutation causes a switch to more aggressive EHE tumours in patients [63].

Only one GEMM investigating YT-driven tumourigenesis has been described to date [11]. This was part of a comparitive study investigating common oncogenic functions across YAP fusion proteins. These included YAP1-MAMLD1 and YAP1-FAM118B found in ependymoma, along with YAP1-SS18, present in cervical squamous cell carcimoma and endocervical adenocarcinoma [64,76,77]. Here, YT expression was introduced using an RCAS retroviral vector, which allowed expression of the fusion proteins only in cells expressing the tv-a receptor [11]. Tv-a receptor expression was under the control of either the GFAP or Nestin promoter. In this model, mice showed both intercranial and muscular tumour formation upon YT expression; an effect which was exasperated in *Cdkn2a*-null mice [11]. Transcriptomic analysis of these tumours demonstrated that YT initiated a similar transcriptional programme to activated YAP, consistant with TC and previous in vitro studies investigating YT [10,11,38]. Additionally, this was a common feature of all the YAP fusion proteins investigated [9,11,24].

### 4.4. Development of an EHE Cell Line

The development of an EHE cell line is important, as it would encompass the benefits of the cell line-based models while being in the correct cellular context. To date, attempts to derive an EHE cell line by expanding cells from tumour samples have proven unsuccessful. One group has been able to generate three EHE cell lines of murine origin, by ex vivo expansion of tumours generated by the GEMM described above [61,62] (Figure 3). This allowed three cell lines to be developed, two of which were homozygous for the *Cdkn2a* deletion, and one which was heterozygous [62]. Ex vivo expansion of GEMM-derived EHE tumours with wild-type *Cdkn2a* was not possible.

These novel EHE cell lines were validated to confirm their relevance to the human disease and findings from previously described model systems [62]. RNA-sequencing of the EHE cell lines showed significant enrichment for canonical TAZ/YAP targets and the EHE gene set [10,38,60,61,62]. EHE cell lines were also able to form tumours upon xenograft into mice that had the histological features of human EHE and produced liver and lung metastases [62]. Additionally, this seminal study of the EHE cell lines demonstrated their usefulness for drug testing [62]. This included the efficacy of trametinib for reducing cell growth, consistant with previous results in TC-expressing HEK293 cells [60]. It was also shown that EHE cell lines are reliant on TC expression to maintain proliferation, although the pharmacological inhibition of TEADs did not lead to a significant reduction in cell viability at lower concentrations [62]. Combining TEAD and MAPK inhibition did, however, lead to a greater reduction in viability of EHE cell lines than either inhibitor alone [62].

## 5. Conclusions and Future Directions

Recent advances mean that there are now a range of model systems available for investigating EHE (Table 1). These utilise different approaches, each with benefits, which complement each other to permit study of various aspects of EHE development (Table 2). This includes multiple in vitro models and three EHE cell lines, which has ethical implications in reducing the number of animals used in research. Many of these models have yet to be used to their full potential, and will form the basis for many future studies, particularly in determining clinically relevant drug targets. Furthermore, these models could be altered in the future to broaden our knowledge of EHE biology and better replicate the heterogeneity of the disease.

Most current studies have focussed on characterising TC function and regulation in EHE development. While this is understandable, as TC is the most common driver fusion protein, future investigation should expand this to include YT-driven EHE, and the rarer TAZ fusion proteins present in cardiac EHEs. The currently available models could utilise the same methods to express the alternative fusion proteins, which would allow comparative studies. This would build on previous studies comparing the transcriptome and chromatin landscape of TC and YT expressing cells, and YT to other YAP fusion proteins [9,10,11,24]. More in-depth comparative studies may reveal causal mechanisms behind the differing survival rates between TC and YT-driven EHE, and will broaden our understanding of the heterogeneous nature of the disease [13]. Moreover, this could reveal drug targets that are efficacious across TAZ/YAP fusion protein-driven malignancies. One example is the use of TEAD-specific inhibitors, which have been examined in the EHE cell lines [62,78]. Altering current model systems could also extend to determining the contribution of secondary mutations besides *CDKN2A* to EHE development, as it is conceivable that these would affect tumour growth to varying degrees. One way to achieve this is to use the CRISPR/Cas9-mediated approach adopted to study *CDKN2A* deletion in the stem cell-based model [63]. This would constitute a more personalised medicine approach and would likely lead to better treatment options for patients.

One previously unexplored area of EHE biology is the contribution of the tumour microenvironment, which would be possible using one of the EHE GEMMs. This would likely uncover novel functions of TC, particularly as wild-type TAZ/YAP has been implicated in crosstalk with the microenvironment [2,79,80,81,82,83,84]. In vitro evidence suggests that TC can be regulated mechanically, favouring a cytoplasmic localisation when actin polymerisation is inhibited [39]. The endothelium is exposed to mechanical signals by the shear stress of blood flow and is known to regulate wild-type TAZ/YAP activity, and thus further study is warranted to determine the contribution of the microenvironment to EHE tumour initiation [85]. Single cell RNA-seq analysis of GEMM-derived EHE tumours also revealed immune cell infiltration, including subsets of B cells, T cells and myeloid cells [61]. In pancreatic and prostate cancers, TAZ/YAP activity has been implicated in tumour-promoting immune cell recruitment, due to TEAD-mediated expression of chemokines and cytokines [80,81,82]. This could represent a previously unexplored mechanism of TC action, which may contribute to EHE development, and may have therapeutic relevance. Together, this evidence how the current EHE model systems can be adapted or utilised in different ways to improve our understanding of the disease. Ultimately, this will lead to better treatment options for patients.

## Figures and Tables

**Figure 1 cancers-15-03005-f001:**
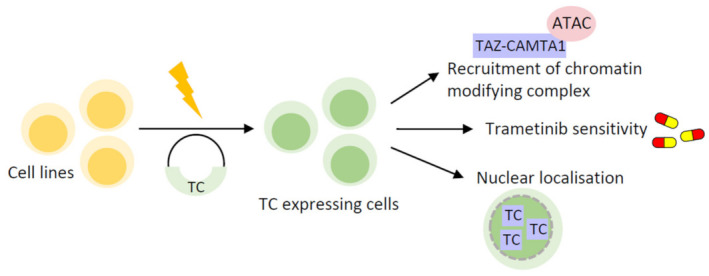
Cell line-based EHE model systems. Schematic showing an example workflow for introducing TC expression into mammalian cell lines, by transfection of a TC expression vector. Key findings from functional studies of TC in cell line-based models of EHE include its ability to bind the ATAC complex, which is also true of YT; the sensitivity of TC expressing cells to the MAPK inhibitor trametinib, both in vitro and in xenograft models in vivo; and the predominantly nuclear localisation of TC, despite LATS1/2 phosphorylation. TC: TAZ-CAMTA1; YT: YAP1-TFE3; ATAC: Ada2a-containing; MAPK: mitogen-activated protein kinase.

**Figure 2 cancers-15-03005-f002:**
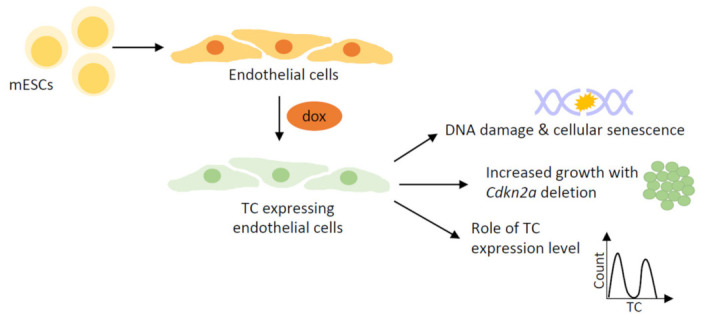
Stem cell-based EHE model systems. Schematic showing the workflow for generating primary endothelial cells from mESCs, with TC expression subsequently induced by dox induction, for generation of stem cell-based models of EHE. Key findings from stem-cell-based modelling of EHE include the generation of DNA damage and subsequent senescence upon TC expression; increased cell growth upon deletion of *Cdkn2a*; and the differences between endothelial populations expressing different levels of TC, with TC high cells often displaying a more severe phenotype. mESCs; mouse embryonic stem cells, dox; doxycycline, TC; TAZ-CAMTA1.

**Figure 3 cancers-15-03005-f003:**
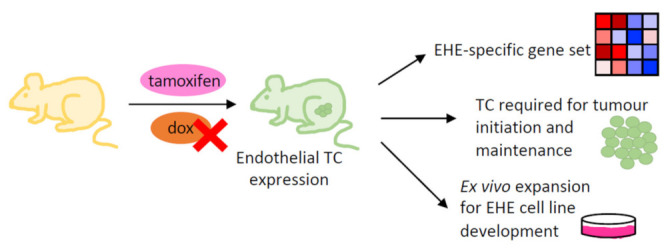
Genetically engineered mouse models of EHE. Schematic showing the development of EHE GEMMs, using either the FLEx system induced by tamoxifen, whereby TC expression is controlled by the *WWTR1* promoter, or a Tet-off system whereby TC expression is repressed by dox administration. Both examples result in endothelial specific TC expression. Key discoveries from EHE GEMMs include the characterisation of and EHE specific gene set, which displays significant overlap with human disease; increased tumour growth and Ki67 expression upon *Cdkn2a* deletion; and the generation of EHE cell lines by ex vivo expansion of GEMM-derived EHE tumours, where *Cdkn2a* has been knocked out. GEMM; genetically engineered mouse model, dox; doxycycline, TC; TAZ-CAMTA1, FLEx; flip-excision system, WWTR1; WW domain containing transcription regulator 1.

**Table 1 cancers-15-03005-t001:** Currently available model systems for studying EHE.

Type	Cell Lines	Mouse Model	Fusion	Reference
**In vitro**	HEK293, NIH3T3	-	TC	Tanas [38]
**In vitro and in vivo**	HEK293	GEMM	YT	Szulzewsky [11]
**In vitro and in vivo**	NIH3T3, SW872	Xenograft	TC/YT	Merritt [10]
**In vitro and in vivo**	MS1	Xenograft and GEMM	TC	Driskill [39]
**In vitro and in vivo**	NIH3T3	Xenograft	TC	Ma [60]
**In vivo**	-	GEMM	TC	Seavey [61]
**In vitro and in vivo**	EHE cell line	GEMM	TC	Seavey [62]
**In vitro**	mESCs	-	TC	Neil [63]

**Table 2 cancers-15-03005-t002:** Key benefits of current approaches to EHE model development.

	Cell Line-Based	Stem Cell-Based	GEMM	EHE Cell Line
Drug testing	✓	-	✓	✓
Generate large amount of sample	✓	✓	-	✓
Reduce the use of animals	-	✓	-	✓
Transfer to in vivo by xenograft	✓	-	-	✓
Correct cell context	-	✓	-	✓
Easy to study other driver fusion proteins	✓	-	-	-
Easy to establish and simple culture conditions	✓	-	-	-
Representative of disease course and tumour histology	-	-	✓	-
Study initiating events	-	✓	-	-
Study TC at various stages of development	-	✓	-	-
Study metastasis	-	-	✓	-
Determine tumour microenvironmental contributions	-	-	✓	-
EHE cell line development	-	-	✓	-

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
