# Peer review of "Current Model Systems for Investigating Epithelioid Haemangioendothelioma"

_cancers, 2023, doi:10.3390/cancers15113005_

Round 1
Reviewer 1 Report
Overall, this is a well-written, comprehensive review article focused on epithelioid hemangioendothelioma and the various model systems that have been used to gain insight into the pathogenesis of this cancer. Only minor revisions requested:
1. Please include full nomenclature for the chromosomal translocations in EHE-- e.g. t(1;3)(p36;q25) and t(X;11) (p11;q22)
2. Please provide more detail regarding the structure of the TAZ-CAMTA1 and YAP-TFE3 fusion proteins (e.g. TAZ and YAP are the N terminal proteins, CAMTA1 and TFE3 are transcription factors and the C terminal proteins, TAZ and YAP are fused in frame to CAMTA1 and TFE3, respectively).
Author Response
We thank the reviewers for their positive comments and suggestions to improve our manuscript. All changes to the manuscript are highlighted in red.
Reviewer 1:
Overall, this is a well-written, comprehensive review article focused on epithelioid hemangioendothelioma and the various model systems that have been used to gain insight into the pathogenesis of this cancer. Only minor revisions are requested:
- Please include full nomenclature for the chromosomal translocations in EHE-- e.g. t(1;3)(p36;q25) and t(X;11) (p11;q22)
This has been added to the manuscript.
- Please provide more detail regarding the structure of the TAZ-CAMTA1 and YAP-TFE3 fusion proteins (e.g., TAZ and YAP are the N terminal proteins, CAMTA1 and TFE3 are transcription factors and the C terminal proteins, TAZ and YAP are fused in frame to CAMTA1 and TFE3, respectively).
This has been added to the manuscript.

Reviewer 2 Report
I read this paper with enthusiasm
It is very well written and understandable
References are precise
I feel it is worth publication
Author Response
We thank the reviewers for their positive comments and suggestions to improve our manuscript. All changes to the manuscript are highlighted in red.
Reviewer 3 Report
The review article provided updated information on model systems for epithelioid hemangioendothelioma. This article is well written, the logic is coherent. And the description is concise and clear.
Page numbers are missing in some references.
Author Response
We thank the reviewers for their positive comments and suggestions to improve our manuscript. All changes to the manuscript are highlighted in red.
Reviewer 3:
The review article provided updated information on model systems for epithelioid hemangioendothelioma. This article is well-written, the logic is coherent. And the description is concise and clear.
Page numbers are missing in some references.
All references have been checked and updated if required.